# Nutritional Quality of the Most Consumed Varieties of Raw and Cooked Rice in Spain Submitted to an In Vitro Digestion Model

**DOI:** 10.3390/foods10112584

**Published:** 2021-10-26

**Authors:** José Raúl Aguilera-Velázquez, Pilar Carbonero-Aguilar, Irene Martín-Carrasco, María Gracia Hinojosa, Isabel Moreno, Juan Bautista

**Affiliations:** 1Department of Biochemistry and Molecular Biology, University of Seville, 41012 Sevill, Spain; jraguilera@us.es (J.R.A.-V.); jdbaut@us.es (J.B.); 2Area of Toxicology, Department of Nutrition and Bromatology, Toxicology and Legal Medicine, University of Seville, 41012 Seville, Spain; imcarrasco@us.es (I.M.-C.); mhinojosa1@us.es (M.G.H.); imoreno@us.es (I.M.)

**Keywords:** rice, heavy metal, ICP-OES, prebiotic, hazard quotient, lifetime cancer risk, bioaccumulation

## Abstract

Rice is one of the most consumed staple foods around the world and its trade is highly globalized. Increased environmental pollution generates a large amount of waste that, in many cases, is discarded close to culture fields. Some species are able to bioaccumulate toxic substances, such as metals, that could be transferred to the food chain. The main goal of this study was to evaluate the content of metallic (Al, Cd, Pb, and Cr) and metalloid elements (As) in 14 of the most consumed varieties of rice in Spain and their effects on human health. The samples were cooked, and human digestion was simulated by using a standard in vitro digestion method. Metallic and metalloid element levels were analysed by Inductively Coupled Plasma Optical Emission Spectrometry (ICP-OES), previous called microwave digestion. Both the human health risk index, Hazard Quotient, and Lifetime Cancer Risk did not show toxic values in any case. Rice with a higher non-digestible fraction showed a higher liberation of proteins and a lower glycemic index. There were no significant differences in the concentrations of metallic and metalloid elements in cooked rice or in the digestible fraction in all varieties analysed. However, Al concentrations were higher than other metals in all varieties studied due to its global distribution. No relationship has been observed between the digestibility of rice and the bioaccessibility of each metallic and metalloid element. All of the studied rice varieties are healthy food products and its daily consumption is safe. The regular monitoring of metals and As in rice consumed in Spain may contribute to improvements in the human health risk evaluation.

## 1. Introduction

Due to globalization, food consumption is nowadays not limited to the production of each country. Rather, the food consumed can come from any part of the world [1]. Thus, rice consumed in Spain is not limited to that produced in the Albufera of Valencia, the Ebro Delta, or the Marsh of Guadalquivir, as, in addition, a large quantity of rice from abroad is consumed. In this sense, rice is considered a staple food for more than half of the population worldwide as is, for many, their unique source of carbohydrates and proteins. Rice is a product valuable not only for its energetic fuel (carbohydrates and proteins) but also for its content of healthy molecules such as vitamin B1 (thiamine) and vitamin B6 (pyridoxine). In 2019, just over 496 million tons of rice were consumed worldwide [2].

Rice is grown under water-flooded conditions, and it is well known that metals and some toxic substances may be taken up by roots and be accumulating in the grain. Metals and metalloids are found naturally in the environment and/or as a result of human activities. Fossil fuel combustion, mining, industrial processing, and the overuse of chemicals are among many sources that contribute to anthropogenic elemental disperse (Almutairi et al., 2021, Regulatory Toxicology and Pharmacology [3]. Different works show that rice has a higher bio-accumulation capacity of metals and metalloids such as lead (Pb), cadmium (Cd), and arsenic (As) from soils compared to other cereals [4]. Chronic exposure to these metals and metalloids can generate serious health problems that comprise a wide spectrum of diseases from allergies to cancer [5,6,7,8]. Long-term consumption of Cd could induce renal failure, interference with the absorption of Ca, as well as the development of some kinds of cancer [9]. Long-term consumption of As could be the cause of the onset of various illnesses such as cancer, skin diseases, cardiovascular diseases, even some kinds of diabetes [10,11,12]. Aluminium (Al) is a widely occurring element in the environment, and it is the third most abundant element in the earth’s crust. Diet is the most important source of Al exposure for non-occupational population. The WHO/FAO JECFA has re-evaluated the safety of this element and lowered the provisional tolerable weekly intake (PTWI) by sevenfold to 1 mg kg^−1^ body weight (bw) in 2007 because of the potential of Al to affect the reproductive and developing nervous system in experimental animals [13,14]. Chronic exposure to Pb has been associated with adverse effects in humans causing diseases such as anemia, headaches, convulsions, muscle weakness, ataxia, tremors, and paralysis [15]. Pb and Al are neurotoxic agents, which may be bioaccumulated in the body, causing serious damage in the central nervous system (CNS) such as the Alzheimer’s diseases, as well as digestive diseases [16]. Exposure through ones diet to combinations of other elements (Ba, Co, Cr, Cu, Hg, Mn, Ni, and Fe) bioaccumulated at low concentrations in rice could also lead to the development of serious diseases [17]. Thus, Cr is an important micronutrient, but it relates to several pathologies, including carcinogenicity. Cr (III) is the essential element for metabolism of glucose, fat, and protein, but excessive ingestion could endanger human health. Cr (VI) has been confirmed to be a carcinogenic substance, leading to lung cancer and skin damage, causing intestinal diseases, poor blood, kidney disease, and asthma [18]. Oral intake of Ni can induce allergic contact dermatitis in Ni-sensitive, headaches, gastrointestinal and respiratory manifestations, lung fibrosis, cardiovascular diseases, lung and nasal cancer, and epigenetic effects [15]. Therefore, analyzing metallic, metalloids, and mineral elements concentration in the diet, and in specific types of food such as in rice, is of major concern.

Spain consumed around 778,780 tons of rice in 2020, out of which 210,000 tons were imported [19]. Burma, Thailand, Argentina, Pakistan, Guyana, Portugal, Cambodia, Italy, France, and Uruguay were the main importing countries, with percentages of 43.19, 29.05, 27.91, 23.66, 20.27, 16.90, 13.47, 13.47, 9.5, 6.95, and 3.68%, respectively [20]. Most of these countries are developing countries, which present high levels of environmental pollution. Therefore, the concentration of metals and metalloids in rice grown in these countries could be critical. There are several studies where the metallic composition of rice is studied but to our knowledge none of them are focused on the metallic and metalloid elements’ bioaccessibility to these wide varieties of rice. Thus, the main goals of this study were to investigate: (i) the compositional characterization of the main types of rice consumed in Spain, (ii) analyses in vitro digestion and bioaccessibility of four metallic elements and arsenic as a metalloid in rice; (iii) evaluate and compare the health risk associated to these minerals in raw, cooked, and digested rice.

## 2. Materials and Methods

### 2.1. Reagents and Materials

All the solvents used, which were of analytical grade or higher, were obtained from Panreac (Barcelona, Spain) or Sigma-Aldrich (Madrid, España). Type I water (>18 MΩ cm) was obtained from a Milli-Q water purification system (Millipore, Bedford, MA, USA). All materials used were made of plastic and cleaned by soaking in 20% (*v/v*) HNO_3_ sub-boiling quality for 4 h, rinsing three times with type I water, according to EPA method 200.8 (EPA. 200) and drying in a laminar flow hood; using as blank 1% (*v/v*) HNO_3_. ICP multielement calibration standard solution (26 elements in HNO_3_, 5%) was purchased in Scharlau (Barcelona, Spain). The digestive enzymes, i.e., pepsin (porcine, 367 units per mg solid, measured as TCA-soluble products using hemoglobin as a substrate) and pancreatin (porcine, 8 × USP specifications of amylase, lipase, and protease), were also purchased from Sigma-Aldrich. Salts were purchased from Sigma-Aldrich.

### 2.2. Samples, Pretreatments and Treatment

Fourteen varieties corresponding to the most consumed varieties in Spain have been used in this study. In total, 42 rice samples were prepared in this study (14 varieties × 3 replicates). Rice origin is shown in Table 1.

As a pretreatment, to remove any possible foreign metallic and metalloid contamination in the rice, all samples of raw rice were washed with deionized distilled water before cooking [21] and dried at 65 °C for 48 h. Washed rice was cooked in boiling water at a ratio of 2:1 (*v/v*) for 15 min and cold down at room temperature. Cold cooked rice was then dried at 65 °C for 48 h.

A sample of 2.5 g of each cooked rice was digested following the standard in vitro digestion method described by Brodkorb et al. [22], simulating human digestion (stock solutions used for simulated digestions are shown in Table 2). Briefly, the method consists of an in vitro simulation of the digestion process, clearly differentiating oral, stomach, and intestinal digestion in phases. Oral phase: the mastication simulation was carried out in a ceramic mortar to avoid contamination from metallic elements. Simulated salivary fluid (SSF) electrolyte stock solution (adjusted to pH 7) was added at a final ratio rice/SSF of 50:50 (*w*/*v*). Human salivary ∝-amylase (EC 3.2.1.1) and CaCl_2_ were added to obtain a final concentration of 75 U (75 U mL^−1^) and 0.75 mM, respectively. Samples were incubated in a shaker Max Q5000 (Thermo Fisher Scientific Inc., Waltham, MA, USA), at 37 °C and 95 rpm, for 2 min. Gastric phase: five parts of oral bolus were mixed with four parts of simulated gastric fluid (SGF) stock electrolyte solution (adjusted to pH 3) to obtain a final ratio food/SGF of 50:50 (*v/v*). Porcine pepsin (EC 3.4.23.1) and CaCl_2_ were added to achieve 2000 U mL^−1^ and 0.075 mM in the final digestion mixture, respectively. Samples were incubated in a shaker Max Q5000 (Thermo Fisher Scientific Inc., Waltham, MA, USA), at 37 °C and 95 rpm, for 2 h. Intestinal phase: Five parts of gastric-chime was mixed with four parts of simulated intestinal fluid (SIF) electrolyte stock solution to obtain (adjusted to pH 7) a final ratio gastric-chime/SIF of 50:50 (*v/v*). CaCl_2_ was added to the gastric-chime mixture, to reach a final concentration of 0.3 mM, followed by the addition of 0.25 mL of a mixture of bile extract and pancreatin (trypsin activity 100 U mL^−1^). Samples were incubated in a shaker Max Q5000 (Thermo Fisher Scientific Inc., Waltham, MA, USA), at 37 °C and 100 rpm, for 2 h. After digestion, the digest was centrifuged at 30.000× *g* for 2 h, obtaining an aqueous-micellar phase (digestible fraction) and a pellet (the non-digestible fraction).

### 2.3. Bioaccessibility

The estimation of the fraction of an ingested biocomponent (metallic and metalloid elements in our case) that becomes accessible for absorption through the epithelial layer of the gastrointestinal tract (GIT) or bioaccessibility was estimated by Equation (1).
(1)$${\mathrm{Bioaccessibility}}_{\mathrm{Metals}}=\frac{\mathrm{mg}\;\mathrm{metal}\;\mathrm{in}\;\mathrm{digestible}\;\mathrm{fraction}}{\mathrm{mg}\;\mathrm{metal}\;\mathrm{in}\;\mathrm{cooked}\;\mathrm{rice}\;}\times 100$$

### 2.4. Physic-Chemical Characterization

Physic-chemical characterization was carried out in raw, cooked, and digested rice. Moisture content was determinate by a gravimetric method; by drying samples at 100 °C until constant weight (approximately 48 h). The total carbohydrates were determined by AOAC 985.29 [23]. Crude proteins were determined by Kjendahl method [24] and fat content was obtained by Soxhlet extraction method [25]. Ash content was determined by gravimetric measurement of the residues of the samples after ignition in a muffle oven at 600 °C during 16 h.

### 2.5. Metal Determination

#### 2.5.1. Sample Treatment

In total, 84 samples (14 varieties of cooked and 14 digested × 3 replicates) were analyzed. One g of each sample was transferred to Teflon vessels (PTFE), which were previously washed with an HNO_3_ solution and Milli-Q water. Next, 4 mL of 65% HNO_3_ were added and 2 mL H_2_O_2_ were added to the samples and the closed Teflon vessels were placed in a microwave digester system (Ethos One, Milestone) for digestion. Digestion conditions were those described by Rubio-Armendariz et al. [26].

#### 2.5.2. Analytical Method

Metallic and metalloid elements were determined by ICP-OES in a Spectro-Blue (Spectro). The conditions of the method are summarized in Table 3. These parameters were established and optimized for a liquid matrix previously by Gutierrez et al. [27].

### 2.6. Calculations

#### 2.6.1. Estimated Daily Intake of Metallic and Metalloid Elements

The estimated daily intake (EDI) of metallic and metalloid elements of rice consumption was calculated according to Equation (2):(2)$$\mathrm{EDI}=\frac{\left(\mathrm{EF}\times \mathrm{ED}\times \mathrm{FIR}\times \mathrm{MC}\right)}{\left(\mathrm{BW}\times \mathrm{AT}\right)}$$
where, EDI corresponds to the estimated daily intake (mg/kg∙d), EF is the exposure frequency (365 d/year), ED is the exposition daily for an adult (54 years), FIR is the food ingestion rate (7.67 g/person∙d), MC indicates the metallic or metalloid element concentration in rice (mg/g dry weight), BW is the mean body weight of an Spanish consumer (68.5 kg) [24] and AT is adult time (d) of 54 years (54 years × 365 d) for non-carcinogenic effects and 70 years (70 years × 365 d) to carcinogenic effects [28].

#### 2.6.2. Non-Carcinogenic Risk Assessment

Non-carcinogenic risk was evaluated for Al, Cr, As, Cd and Pb through Target Hazard Quotient (THQ) method described by Chien et al., [29]. Equation (3) is used to calculate the THQ is shown below.
(3)$$\mathrm{THQ}=\frac{\mathrm{EDI}}{\mathrm{Rfd}}$$
where, Rfd is the oral reference dose (mg/kg∙d), and EDI, the estimated daily intake as is described above. The Rfd of each metallic or metalloid element [30] is shown in Table 4.

The Total Target Hazard Quotient (TTHQ) was calculated using Equation (4), by adding up the THQ of all metallic and metalloid elements. Values below 1 in THQ and TTHQ indicate that the risk is acceptable for chronic systemic effects, and values above 1 show an unacceptable non-carcinogenic risk.
(4)$$\mathrm{TTHQ}={\mathrm{THQ}}_{\mathrm{Al}}+{\mathrm{THQ}}_{\mathrm{Cr}}+{\mathrm{THQ}}_{\mathrm{As}}+{\mathrm{THQ}}_{\mathrm{Cd}}+{\mathrm{THQ}}_{\mathrm{Pb}}$$

#### 2.6.3. Carcinogenic Risk Assessment (CRA)

CRA was estimated for As, Cd, Cr and Pb, which allowed to obtain the incremental lifetime cancer risk (ILCR), according to Equation (5), where CSF is the cancer slope factor, which are indicated for each metallic and metalloid element in Table 5. Regarding to USEPA criteria, a range from 10^−6^ to 10^−4^ is considered as a tolerable carcinogenic risk, and values above 10^−4^ are regarded as intolerable to population [31].
(5)ILCR=EDI×CSF

In both cases, the non-carcinogenic and carcinogenic health risk of digestible fractions were estimated by subtracting the non-digestible fraction values from the cooked rice values.

### 2.7. Statistical Analyses

The IBM SPSS statistic software has been used to statistical analysis. One-way variance (ANOVA) was used to make statistic comparison between the concentration of each metallic and metalloid element in both rice processed, cooked, and digested, with a t-test; *p* < 0.05 was considered significant. Kolmogorov-Smirnov was used to verify the normality and Levene’s test to analyse the homogeneity of variance based on the mean.

## 3. Result and Discussion

### 3.1. Chemical Characterization of Raw Rice

The diversity of the geographical origins of rice consumed in Spain can play an important role on its chemical composition, not only in the content of metallic or metalloid elements, thus, the chemical composition of the 14 major varieties of rice consumed in Spain was analyzed. Results for major components are shown in Table 6. In all varieties the content of macro-components: carbohydrates, proteins, ash, and fats are within normal range for rice, with no significant differences, and similar to that described by others [32].

### 3.2. Chemical Characterization of Cooked Rice

As rice is often eaten cooked, its physic-chemistry characterization was also carried out. Results expressed on a dry weight basis are shown in Table 7. The main difference between both forms, raw and cooked, is in the moisture content, for which significant differences were found (average values: 12.62% ± 1.08 and 4.09% ± 0.52, respectively, *p* < 0.001), understanding that this moisture is the capacity of rice to keep water after the drying process. These differences may be due to gelatinization processes, which take place during the cooking and cooling process [33]. The structure of the complex gel system and its water retention capacity depend on intra- and inter-carbohydrate association by hydrogen bonds or Van der Waals forces. Water is retained back stronger by starch in raw rice due to its crystalline structure (granule), than in cooked rice with a gel structure (much more compact). This is due to the process named syneresis, where water is wept, caused by a prolonged cooling of cooked rice [34]. During the cooling process the amylose-amylose and amylose-amylopectin hydrogen bonds give rise to squeezing out of the water between these polymers. In relation to individual data, as shown in Figure 1 and Table 6 and Table 7; in raw rice moisture values ranges between 14.1% ± 1.5 and 10.8% ± 1.0 found in the varieties Memby and Piñana, respectively, with an average moisture of 12.62% ± 1.08, These results validate the ones previously described by Lim et al. in 2003 and Ozbekova in 2019 [35,36]. In cooked rice moisture is much lower, showing an average of 4.09% ± 0.52, ranging between 5.2% ± 6.3 and 3.1% ± 4.2, found in the varieties Perlado and Thaiperla, respectively. However, it has not been possible to find preceding works that measured cooked rice moisture content by drying the rice before measuring. Indeed, values available in bibliography appear to be between 100–120%, for example in this study carried out by Wu et al. in 2017 [37].

The concentration of the main components (proteins, carbohydrates, etc.) are higher in cooked rice than in raw rice, as a consequence of the significant differences in moisture between them, as discussed above. However, all data obtained were similar to those previously described by Vini et al. with a mean protein value of 7.25 ± 0.02%, and a mean carbohydrates value of 76.44 ± 0.03% [38]. Indeed, the data are aligned with the results previously achieved by Yankah et al., with protein values of 7.89 ± 0.16% and carbohydrate of 74.09 ± 0.41% [39].

### 3.3. Digestible and Non-Digestible Fractions

In vitro digestion of cooked rice was carried out by following a standard in vitro digestion method previously described by Brodkorb et al. [22]. This method allows obtaining two fractions: one digestible, available to be absorbed through the intestinal epithelium, and a non-digestible fraction, which passes directly to the colon and can be used as prebiotic [40] and/or be eliminated. The digestion process in cooked rice was performed to evaluate which proportion of its components is able to be assimilated and excreted.

As shown in Figure 2, the content of non-digestible and digestible fraction in cooked rice shows significant differences. Non-digestible fraction ranged between 25.5% and 73.1%, found in the varieties Thaiperla and Guadiamar, respectively, with an average value of 44.3% ± 12.9. These differences could be explained due to the presence of resistant starches (non-hydrolysable starches and slow digestibility starches) mainly. The high content of non-digestible and/or slow-digestibility components makes them good candidates for diabetes and obesity treatments [41], due to their lower glycemic index and their possible effect as prebiotics [42].

Protein content in the digestible fraction was calculated as the difference between total proteins in cooked rice (Table 7) and proteins in non-digestible fraction (Table 8). These results show that protein content in cooked rice oscillated between 15.49% ± 1.4 and 6.59% ± 0.5 in Barone and Thaiperla varieties, respectively, with an average value of 10.37% ± 2.2, which is similar to the mean values previously described by Rezvan et al. (9.48 ± 0.04 and 8.87 ± 0.04%) in cooked Hashemi and Domsiyah varieties and the 9.74 ± 0.21% found in cooked MGS variety by Kulwa et al. [43,44]. While in the digestible fraction, protein content has an average value of 8.15% ± 0.40, oscillating its values between 12.87% ± 1.73 and 4.57% ± 0.27 in Basmati and Jasmine varieties, respectively. This difference could be due to different ways of protein storage in the grain. In rice, there are two structures to store proteins, body proteins and vacuoles. These structures lend different degrees of solubility of proteins [40] and both structures may be modified by the cooking process. That could explain the differences in the content of proteins between digested and cooked rice.

The estimation of the amount of assimilated protein in reference to consumed rice was calculated from these results (see Figure 3). These data show that the average value of for protein liberation is 4.84% ± 2.65 and that the highest and lower values (8.71 ± 0.51 and 1.65 ± 0.4 g/100 g of cooked rice) were observed for the varieties Basmati and Guadiamar, respectively.

According to these results, it can be assumed that those varieties with a larger digestible fraction provide a higher amount of protein. In addition, it seems reasonable to assume that highly digestible varieties will present a greater release of metallic and metalloid elements than varieties with lower digestibility. To confirm this hypothesis, metallic elements and As content were analyzed in all the rice varieties studied in this work.

### 3.4. Metalllic and Metalloid Content in Rice

In addition to the differences found in the composition of macro-components (carbohydrates, Proteins, etc.), the composition of micro-components, mainly metals, could also show significant differences. Therefore, our study was focused on the content of metallic and metalloid elements, analyzing them after being cooked and digested as an indirect measure of their bioaccessibility. Results are shown in Table 9A,B. The highest levels were those provided by Al in all the varieties, which is not surprising due to its global distribution [45]. Aluminum content ranged between 1.9 × 10^−3^ ng/g and 3.1 ng/g in Memby and Piñana varieties, respectively. The average value was of 2.5 ± 0.6 × 10^−3^ ng/g in cooked rice, which are similar values than those described by Zhao [32]. However, the values in the non-digestible fraction were lower, varying between 1.1 × 10^−3^ ng/g and 7.4 × 10^−4^ ng/g in Piñana and Basmati varieties, respectively, with an average value of 1.7 ± 0.6 × 10^−3^ ng/g. These data show that the amount of Al released is not the same in all varieties and that the degree of release depends on the type of matrix of each variety. The highest degree of Al release is observed for the Perlado variety (71.63% ± 8.21). Despite Al is considered as neurotoxic, related with neurodegenerative diseases such as Parkinson and Alzheimer’s, its concentration in the digestible fraction was below the toxic threshold fixed in 1 mg/person/week, 10^5^ fold lower that the toxic threshold [46,47].

Predictably, the Cd, Pb, Cr, and As concentrations were found below toxic limits, 9, 9, 7 and 7 fold below than toxic threshold, respectively [48,49,50,51], including the As values obtained in Basmati variety in cooked rice (increased 10 fold). Concentrations equal to the Limit of Quantification (LOQ) were assumed for Cd, Pb, Cr, and As in the worst scenario.

Metallic elements bioaccessibility is shown in Table 10, depending on the digestibility grade of each rice variety. The highest percentage of bioaccessibility has been found for Al (87.6% ± 10.25) in Sona Masoori variety. In contrast, the lowest percentage of bioaccessibility has been found for Cr (22.4% ± 2.5), also in the same variety. Even though this is the lowest value, it is substantially higher than the ones found by other authors. For instance, Kumari and Platel described a Cr bioaccessibility of 8 ± 0.17% [52].

The presence of metallic and metalloid elements after the digestion process could be explained since they keep embedded into the non-digestible fraction as previously described in rats by Rose et al. [53]. The presence or formation of resistant or low digestibility starches may contribute to establishing a stronger metallic and metalloid element-retention than might be expected to fast digestibility starches. Recent works demonstrated the capacity of some modified starches to form metallic or metalloid elements-complexes [54], thus, a lesser grade these modifications may occur spontaneously in rice, which could explain the higher presence of these metallic and metalloid elements in the non-digestible fraction of rice.

### 3.5. Metal Intake through Rice Consumption

The average intake of all metallic elements/person/d was estimated from the amount of assimilated metallic elements calculated and the Spanish average rice consumption (7.67 g/∙person∙d). It was calculated which percentage correspond of the oral reference values of each metallic element, (2.41–8.54% in Al; 0.02–0.05% in Cd; 0.02–0.85% in Cr; 0.02–0.05% in Pb; 0.04–9.91% in As) (Table 11). All values obtained are far from the toxic threshold to each metallic element.

### 3.6. Human Health Risk Assessment

The human health risk assessment comprises the evaluation of non-carcinogenic and carcinogenic health risks. To assess the level of concern arising from the ingestion of metallic and metalloid elements in rice the THQ was evaluated. THQ is defined as a ratio of exposition to toxic elements and the reference doses, which is the highest level before the appearance of harmful effects. The THQ values for all the elements were below 1; thus, it may be concluded that there is not potential non-carcinogenic health risk. Similar results were previously obtained by Huang et al. [4], who observed the absence of health risk (Cd, Pb, Hg, and As) through rice consumption. However, there are some studies carried out in several countries that describe a THQ exceeding the threshold limit, as it was reported by Hensawang et al. in 2017 in Thailand, Munish et al. in India and Tapos et al. in Bangladesh [55,56,57]. Once THQ in the digestible fraction was estimated (Figure 4), the highest value was observed in the Puntal variety (0.0045), and the lowest in the Barone variety (0.001).

In reference to the carcinogenic health risk, the ILCR (understood as the probability of someone to develop cancer due to the exposure to a carcinogenic element) values obtained in all digested rice were still far from those considered carcinogenic (<1 × 10^−4^) as previously described by USEPA [53]. Most of the varieties of rice, in cooked digested rice, showed similar ILCR values, being all of them around 10^−8^ (Figure 5). However, there were two exceptions. The first one was the Basmati variety (cooked and digested), whereby higher concentrations of As were detected in comparison to the rest of the varieties. This may be explained by the higher contamination levels in India compared to the other countries. Nonetheless, the Sona Masoori variety cultivated in India also showed lower concentrations of As than the Basmati variety, and closer to the rest of the varieties studied. This fact might be related with the non-homogeneous As content in the different cultivated areas in India, since Zavala and Duxbury described that As content in rice depends on soil composition [58]. The second one was the Piñana variety, which Cr levels in the digestible fraction were higher than the rest of the elements [59]. The results of ILCR of each metallic and metalloid element separately did not show any carcinogenic risk, as should be expected for marketed rice. However, in order to have an overall assessment of the carcinogenic risk, combinations of different concentrations of metallic and metalloid elements below threshold carcinogenic risk should be studied.

## 4. Conclusions

In the present study, there were no significant differences in the metallic and metalloid elements-concentration in rice between all the varieties analyzed, which demonstrates that the monitorization of rice consumed carried out in Spain by companies and public health agencies is vital in order to maintain these concentrations below metallic/metalloid-toxic threshold. Moreover, rice with a higher non-digestible fraction showed a higher liberation of proteins and a lower glycemic index. Both factors are features of healthy food. No relationship has been observed between the digestibility of rice and the bioaccessibility of each metallic and metalloid element. These data should be analysed carefully, as rice is usually consumed as a side dish in developed countries and the presence of other foodstuffs could contribute to increasing human health risk.

## Figures and Tables

**Figure 1 foods-10-02584-f001:**
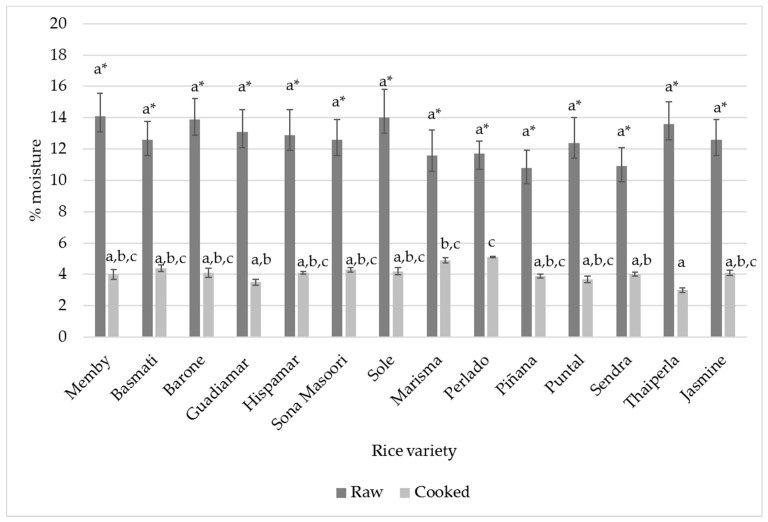
Moisture (%) in raw and cooked rice. Different letters in the same row indicate significant differences by ANOVA test (*p* < 0.05). * indicates significant differences between raw and cooked rice.

**Figure 2 foods-10-02584-f002:**
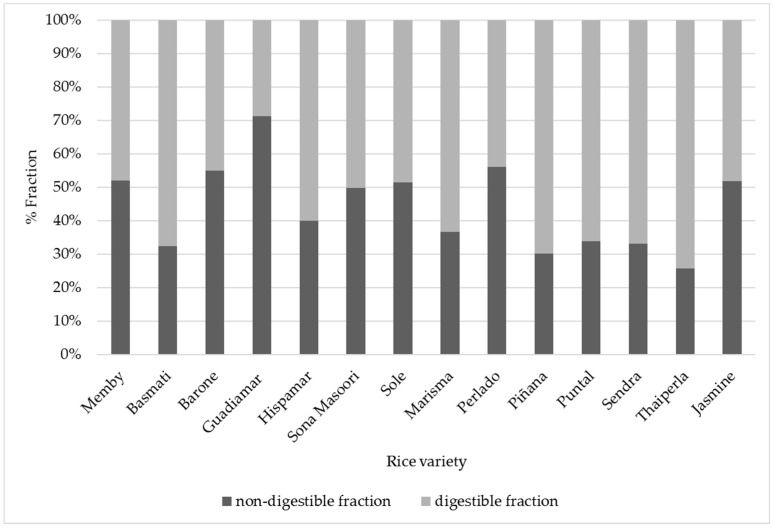
Digestible and non-digestible fraction analysis. Results expressed as %.

**Figure 3 foods-10-02584-f003:**
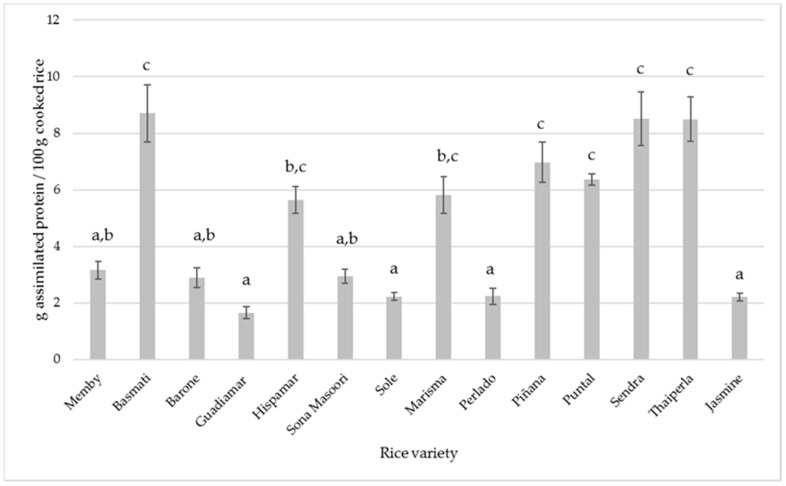
Estimation of assimilated protein in cooked rice consumed (g/100 g). Different letters in the same row indicate significant differences by ANOVA test (*p* < 0.05).

**Figure 4 foods-10-02584-f004:**
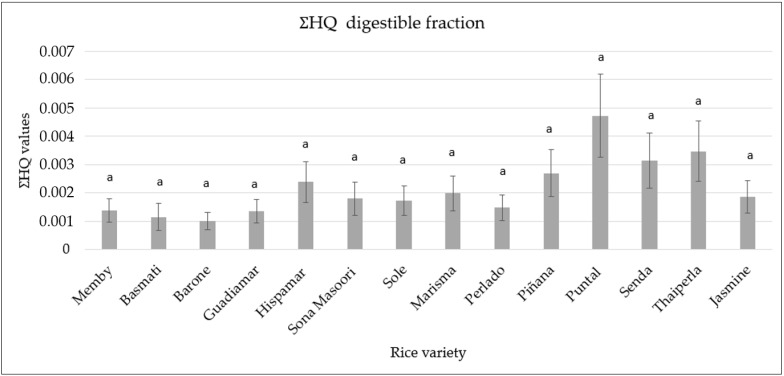
Total Hazard Quotient (∑THQ) of digestible fraction. Different letters in the same row indicate significant differences by ANOVA test (*p* < 0.05).

**Figure 5 foods-10-02584-f005:**
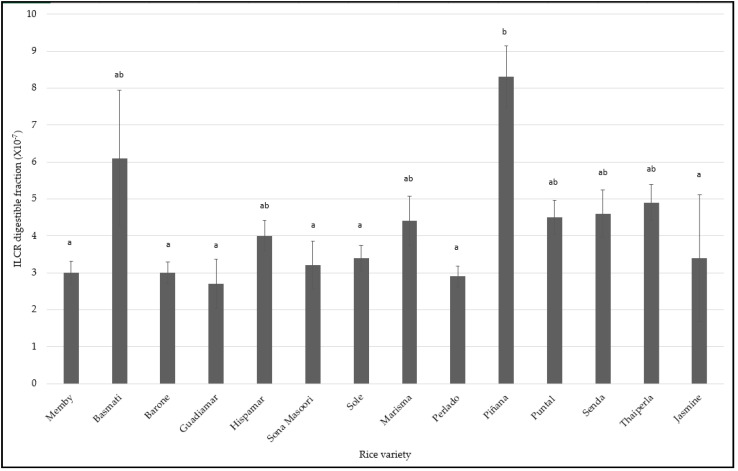
ILCR estimated of digestible fraction. Different letters in the same row indicate significant differences by ANOVA test (*p* < 0.05).

**Table 1 foods-10-02584-t001:** Rice samples from different origins.

Variety	Origin
Hispamar	Spain
Perlado	Spain
Memby	Argentine
Barone	Greece
Jasmine	Vietnam
Sona Masoori	India
Sole	Italy
Piñana	Spain
Thaiperla	Spain
Guadiamar	Spain
Puntal	Spain
Sendra	Spain
Basmati	India
Marisma	Spain

**Table 2 foods-10-02584-t002:** Preparation of stock solutions of simulated digestion fluids. Final volume of 500 mL for each fluid.

			SSF		SGF		SIF	
			pH7		pH3		pH7	
Constituent	Stock Concentration		Volume of Stock	Concentration in SSF	Volume of Stock	Concentration in SSF	Volume of Stock	Concentration in SSF
	g L^−1^	mol L^−1^	mL	mmol L^−1^	mL	mmol L^−1^	mL	mmol L^−1^
KCl	37.3	0.5	15.1	15.1	6.9	6.9	6.8	6.8
KH_2_PO_4_	68	0.5	3.7	3.7	0.9	0.9	0.8	0.8
NaHCO_3_	84	1	6.8	13.6	12.5	25	42.5	85
NaCl	117	2	-	-	11.8	47.2	9.6	38.4
MgCl_2_(H_2_O)_6_	30.5	0.15	0.5	0.15	0.4	0.1	1.1	0.33
(NH_4_)_2_CO_3_	48	0.5	0.06	0.06	0.5	0.5	-	-
For pH adjustment								
	g L^−1^		mL	mmol L^−1^	mL	mmol L^−1^	mL	mmol L^−1^
NaOH	1		-	-	-	-	-	-
HCl	6		0.09	1.1	1.3	15.6	0.7	8.4
	g L^−1^	mol L^−1^		mmol L^−1^		mmol L^−1^		mmol L^−1^
CaCl_2_(H_2_O)_2_	44.1	0.3		1.5 (0.75 *)		0.15 (0.075 *)		0.6 (0.3 *)

* in brackets the corresponding Ca^2+^ concentration in the final digestion mixture SSF: Simulated Salivary Fluid; SGF: Simulated Gastric Fluid; SIF: Simulated Intestinal Fluid. Table adapted from Brodkord et al. (2019).

**Table 3 foods-10-02584-t003:** ICP conditions.

Flow Rate	1.00
Radio frequency power	1.350
Plasma argon flow rate (L min^−1^)	15
Auxiliar argon flow rate (L min^−1^)	0.50
Nebulization flow rate (L min^−1^)	0.60
Read delay/s	30
Integration time/s (min-max)	1–5
Replicate	3
Wavelengths (nm)	Cd: 226.5, Pb: 220.3, Al: 237.7, As:228.0, Cr: 205.5

**Table 4 foods-10-02584-t004:** Oral reference value of metallic or metalloid element (mg/kg∙d).

Al	0.0004
Cr	0.0030
As	0.0003
Cd	0.0010
Pb	0.0035

**Table 5 foods-10-02584-t005:** CSF to each metallic and metalloid element studied described USEPA’s guideline.

As	1.5000
Cd	0.3800
Cr	0.5000
Pb	0.0085

**Table 6 foods-10-02584-t006:** Chemical composition of 14 major varieties of raw rice consumed in Spain.

Variety	Moisture (%)	Ash (%)	Fat (%)	Protein (%)	Total Carbohydrates (%)
**Memby**	14.17 ± 1.56 ^a^	0.64 ± 0.03 ^d^	0.62 ± 0.02 ^b^	9.60 ± 1.33 ^a,b^	75.04 ± 7.66 ^a^
**Basmati**	12.61 ± 0.94 ^a^	0.59 ± 0.06 ^c,d^	0.50 ± 0.06 ^a,b^	11.45 ± 2.34 ^a,b^	74.86 ± 8.14 ^a^
**Barone**	13.93 ± 1.52 ^a^	0.44 ± 0.06 ^a,b,c,d^	0.68 ± 0.08 ^b^	15.09 ± 0.48 ^b^	69.89 ± 5.28 ^a^
**Guadiamar**	13.13 ± 2.14 ^a^	0.39 ± 0.01 ^a,b,c^	0.59 ± 0.03 ^b^	9.71 ± 0.97 ^a,b^	76.21 ± 8.29 ^a^
**Hispamar**	12.94 ± 1.67 ^a^	0.43 ± 0.03 ^a,b,c,d^	0.67 ± 0.02 ^b^	9.36 ± 2.03 ^a,b^	76.64 ± 4.13 ^a^
**Sonamasoori**	17.04 ± 1.63 ^a^	0.30 ± 0.03 ^a,b^	0.22 ± 0.02 ^a^	10.26 ± 1.31 ^a,b^	72.22 ± 4.66 ^a^
**Sole**	13.56 ± 2.15 ^a^	0.52 ± 0.03 ^b,c,d^	0.64 ± 0.04 ^b^	6.85 ± 0.95 ^a^	78.49 ± 8.54 ^a^
**Marisma**	11.64 ± 1.44 ^a^	0.45 ± 0.02 ^a,b,c,d^	0.72 ± 0.09 ^b^	10.03 ± 1.63 ^a,b^	77.20 ± 6.94 ^a^
**Perlado**	10.71 ± 1.13 ^a^	0.43 ± 0.03 ^a,b,c,d^	0.68 ± 0.08 ^b^	5.01 ± 0.21 ^a^	82.17 ± 6.55 ^a^
**Piñana**	10.87 ± 1.07 ^a^	0.37 ± 0.04 ^a,b,c^	0.64 ± 0.04 ^b^	9.16 ± 1.82 ^a,b^	79.03 ± 5.30 ^a^
**Puntal**	12.46 ± 1.49 ^a^	0.40 ± 0.05 ^a,b,c^	0.56 ± 0.06 ^b^	6.27 ± 0.75 ^a^	80.37 ± 6.44 ^a^
**Sendra**	10.95 ± 1.17 ^a^	0.52 ± 0.03 ^b,c,d^	0.61 ± 0.04 ^b^	10.23 ± 1.15 ^a,b^	77.74 ± 6.98 ^a^
**Thaiperla**	13.65 ± 1.42 ^a^	0.48 ± 0.04 ^a,b,c,d^	0.54 ± 0.06 ^a,b^	5.21 ± 0.29 ^a^	80.17 ± 6.57 ^a^
**Jasmine**	12.64 ± 1.63 ^a^	0.26 ± 0.06 ^a^	0.7 ± 0.09 ^b^	9.82 ± 1.26 ^a,b^	76.62 ± 6.26 ^a^
**Average value**	12.62% ± 1.08	0.44 ± 0.3	0.60 ± 0.05	9.14 ± 1.18	76.90 ± 6.55

Different letters in the same row indicate significant differences by ANOVA test (*p* < 0.05).

**Table 7 foods-10-02584-t007:** Chemical composition of 14 major varieties of cooked rice consumed in Spain.

Variety	Moisture (%)	Ash (%)	Fat (%)	Protein (%)	Total Carbohydrates (%)
**Memby**	4.00 ± 0.35 ^a^	0.74 ± 0.02 ^f^	0.52 ± 0.03 ^a,b,c^	10.66 ± 1.15 ^a,b^	84.14 ± 4.56 ^a^
**Basmati**	4.44 ± 0.24 ^a^	0.54 ± 0.03 ^c,d,e^	0.59 ± 0.04 ^b,c,d^	12.31 ± 2.06 ^a,b^	82.16 ± 6.34 ^a^
**Barone**	4.14 ± 0.53 ^a^	0.52 ± 0.05 ^b,c,d,e^	0.62 ± 0.07 ^b,c,d^	15.49 ± 1.41 ^b^	79.27 ± 3.76 ^a^
**Guadiamar**	3.57 ± 0.28 ^a^	0.36 ± 0.02 ^a,b,c^	0.49 ± 0.03 ^a,b,c^	10.97 ± 1.35 ^a,b^	84.68 ± 6.97 ^a^
**Hispamar**	4.13 ± 0.45 ^a^	0.65 ± 0.02 ^e,f^	0.61 ± 0.03 ^b,c,d^	10.98 ± 1.41 ^a,b^	83.66 ± 8.75 ^a^
**Sonamasoori**	4.37 ± 0.34 ^a^	0.41 ± 0.03 ^a,b,c,d^	0.31 ± 0.02 ^a^	11.53 ± 1.13 ^a,b^	83.45 ± 4.63 ^a^
**Sole**	4.26 ± 0.47 ^a^	0.55 ± 0.02^,d,e^	0.59 ± 0.05 ^b,c,d^	7.25 ± 0.68 ^a^	87.41 ± 8.12 ^a^
**Marisma**	4.98 ± 0.55 ^a^	0.51 ± 0.03 ^b,c,d,e^	0.81 ± 0.07 ^d^	10.74 ± 1.64 ^a,b^	83.04 ± 4.21 ^a^
**Perlado**	4.01 ± 0.63 ^a^	0.34 ± 0.03 ^a,b^	0.59 ± 0.06 ^b,c,d^	11.98 ± 1.43 ^a,b^	80.66 ± 8.75 ^a^
**Piñana**	5.15 ± 0.61 ^a^	0.32 ± 0.04 ^a^	0.61 ± 0.03 ^b,c,d^	9.56 ± 1.21 ^a,b^	89.51 ± 3.91 ^a^
**Puntal**	3.96 ± 0.45 ^a^	0.43 ± 0.04 ^a,b,c,d^	0.48 ± 0.04 ^a,b^	7.87 ± 0.63 ^a^	87.32 ± 5.25 ^a^
**Sendra**	3.74 ± 0.36 ^a^	0.62 ± 0.02 ^e^	0.75 ± 0.03 ^c,d^	11.46 ± 0.64 ^a,b^	83.47 ± 7.15 ^a^
**Thaiperla**	4.00 ± 0.29 ^a^	0.41 ± 0.04 ^a,b,c,d^	0.62 ± 0.05 ^b,c,d^	6.59 ± 0.56 ^a^	88.38 ± 6.55 ^a^
**Jasmine**	3.00 ± 0.34 ^a^	0.32 ± 0.05 ^a^	0.64 ± 0.07 ^b,c,d^	10.30 ± 1.67 ^a,b^	85.74 ± 5.34 ^a^
**Average value**	4.09% ± 0.52	0.48 ± 0.03	0.58 ± 0.04	10.33 ± 1.21	84.49 ± 6.02

Different letters in the same row indicate significant differences by ANOVA test (*p* < 0.05).

**Table 8 foods-10-02584-t008:** Percentage of protein in non-digestible and digestible fraction.

Variety	Proteins Measured in Non-Digestible Fraction (%)	Calculated Proteins in Digestible Fraction (%)
**Memby**	8.88 ± 0.81 ^b,c,d,e^	6.59 ± 0.62 ^a,b^
**Basmati**	10.53 ± 1.34 ^d,e^	12.87 ± 1.73 ^c^
**Barone**	10.12 ± 1.23 ^c,d,e^	6.44 ± 0.67 ^a,b^
**Guadiamar**	10.29 ± 0.91 ^d,e^	5.77 ± 1.09 ^a,b^
**Hispamar**	8.87 ± 0.81 ^b,c,d,e^	9.40 ± 1.10 ^a,b,c^
**Sona Masoori**	6.17 ± 0.53 ^a,b,c,d^	5.87 ± 0.82 ^a,b^
**Sole**	4.17 ± 0.36 ^a^	4.59 ± 0.37 ^a^
**Marisma**	6.93 ± 0.54 ^a,b,c,d,e^	9.19 ± 1.00 ^a,b,c^
**Perlado**	6.74 ± 0.62 ^a,b,c,d^	5.08 ± 0.40 ^a^
**Piñana**	5.58 ± 0.58 ^a,b,c^	9.99 ± 1.10 ^a,b,c^
**Puntal**	6.42 ± 0.45 ^a,b,c,d,e^	9.61 ± 1.34 ^a,b,c^
**Sendra**	10.88 ± 1.34	12.75 ± 1.09 ^c^
**Thaiperla**	5.44 ± 0.66 ^a,b,c^	11.43 ± 1.34 ^b,c^
**Jasmine**	4.20 ± 0.45 ^a,b^	4.58 ± 0.27 ^a^

Different letters in the same row indicate significant differences by ANOVA test (*p* < 0.05).

**Table 9 foods-10-02584-t009:** (**A**). Metallic and metalloid elements values in cooked rice; (**B**). Metallic and metalloid elements values in the digestible fraction of rice.

(**A**)					
**ng/g of Cooked Rice**					
	**Al**	**Cd**	**Cr**	**Pb**	**As**
**Barone**	2.8 × 10^−3^ ± 3.1 × 10^−4 b^	1.8 × 10^−5^ ± 2.5 × 10^−7 a^	5.2 × 10^−5^ ± 6.8 × 10^−7 c,d,e^	1.8 × 10^−5^ ± 1.7 × 10^−6 a^	3.6 × 10^−5^ ± 3.5 × 10^−6 a^
**Basmati**	2.1 × 10^−3^ ± 2.5 × 10^−4 a,b^	1.8 × 10^−5^ ± 2.3 × 10^−7 a^	6.1 × 10^−5^ ± 4.9 × 10^−6 d,e^	1.8 × 10^−5^ ± 1.8 × 10^−6 a^	3.2 × 10^−5^ ± 1.0 × 10^−6 b^
**Guadiamar**	2.3 × 10^−3^ ± 2.2 × 10^−4 a,b^	1.8 × 10^−5^ ± 1.6 × 10^−6 a^	3.2 × 10^−5^ ± 4.8 × 10^−7 b,c^	1.8 × 10^−5^ ± 2.2 × 10^−6 a^	3.6 × 10^−5^ ± 3.2 × 10^−6 a^
**Hispamar**	2.2 × 10^−3^ ± 2.1 × 10^−4 a,b^	1.8 × 10^−5^ ± 1.8 × 10^−6 a^	2.1 × 10^−5^ ± 1.9 × 10^−6 a,b^	1.8 × 10^−5^ ± 2.7 × 10^−6 a^	3.6 × 10^−5^ ± 4.0 × 10^−6 a^
**Jasmine**	2.3 × 10^−3^ ± 2.3 × 10^−5 a,b^	1.8 × 10^−5^ ± 3.2 × 10^−13 a^	6.5 × 10^−5^ ± 6.5 × 10^−8 e^	1.8 × 10^−5^ ± 2.3 × 10^−6 a^	3.6 × 10^−5^ ± 5.8 × 10^−6 a^
**Marisma**	1.9 × 10^−3^ ± 2.3 × 10^−5 a^	1.8 × 10^−5^ ± 1.6 × 10^−6 a^	3. × 10^−5^ ± 4.3 × 10^−7 b,c^	1.8 × 10^−5^ ± 1.4 × 10^−6 a^	3.6 × 10^−5^ ± 3.5 × 10^−6 a^
**Memby**	1.9 × 10^−3^ ± 2.9 × 10^−5 a^	1.8 × 10^−5^ ± 2.0 × 10^−6 a^	8.7 × 10^−6^ ± 1.1 × 10^−7 a^	1.8 × 10^−5^ ± 2.7 × 10^−7 a^	3.6 × 10^−5^ ± 4.5 × 10^−6 a^
**Perlado**	2.6 × 10^−3^ ± 3.4 × 10^−5 a,b^	1.8 × 10^−5^ ± 2.9 × 10^−6 a^	1.6 × 10^−4^ ± 2.1 × 10^−6 f^	1.8 × 10^−5^ ± 1.6 × 10^−6 a^	3.6 × 10^−5^ ± 4.9 × 10^−6 a^
**Piñana**	2.3 × 10^−3^ ± 1.8 × 10^−4 a,b^	1.8 × 10^−5^ ± 1.8 × 10^−6 a^	4.7 × 10^−5^ ± 3.8 × 10^−6 c,d,e^	1.8 × 10^−5^ ± 1.8 × 10^−8 a^	3.6 × 10^−5^ ± 3.0 × 10^−6 a^
**Puntal**	3.1 × 10^−3^ ± 4.7 × 10^−5 a^	1.8 × 10^−5^ ± 2.3 × 10^−6 a^	6.3 × 10^−5^ ± 7.6 × 10^−6 d,e^	1.8 × 10^−5^ ± 2.2 × 10^−7 a^	3.6 × 10^−5^ ± 2.8 × 10^−6 a^
**Sendra**	2.5 × 10^−3^ ± 2.2 × 10^−4 a,b^	1.8 × 10^−5^ ± 2.4 × 10^−6 a^	6.0 × 10^−5^ ± 5.7 × 10^−6 d,e^	1.8 × 10^−5^ ± 2.3 × 10^−7 a^	3.6 × 10^−5^ ± 3.6 × 10^−6 a^
**Sole**	2.2 × 10^−3^ ± 2.2 × 10^−6 a,b^	1.8 × 10^−5^ ± 1.5 × 10^−6 a^	6.0 × 10^−5^ ± 5.6 × 10^−6 d,e^	1.8 × 10^−5^ ± 2.3 × 10^−7 a^	3.6 × 10^−5^ ± 5.8 × 10^−6 a^
**Sona Masoori**	2.3 × 10^−3^ ± 2.8 × 10^−5 a,b^	1.8 × 10^−5^ ± 1.4 × 10^−6 a^	6.7 × 10^−5^ ± 6.7 × 10^−7 d,e^	1.8 × 10^−5^ ± 1.4 × 10^−6 a^	3.6 × 10^−5^ ± 3.5 × 10^−6 a^
**Thaiperla**	2.4 × 10^−3^ ± 3.1 × 10^−5 a,b^	1.8 × 10^−5^ ± 1.8 × 10^−6 a^	4.3 × 10^−5^ ± 3.9 × 10^−6 c,d^	1.8 × 10^−5^ ± 2.7 × 10^−7 a^	3.6 × 10^−5^ ± 4.5 × 10^−6 a^
(**B**)					
**ng/g of Digestible Fraction of Rice**					
	**Al**	**Cd**	**Cr**	**Pb**	**As**
**Barone**	1.3 × 10^−3^ ± 1.7 × 10^−9^^a,b,c,d^	8.6 × 10^−6^ ± 1.0 × 10^−7 b,c^	1.8 × 10^−5^ ± 1.6 × 10^−6 a^	8.6 × 10^−5^ ± 7.8 × 10^−7 a,b^	1.7 × 10^−5^ ± 2.4 × 10^−7 a^
**Basmati**	2.2 × 10^−3^ ± 4.8 × 10^−9 f^	1.2 × 10^−5^ ± 1.8 × 10^−7 d^	1.8 × 10^−5^ ± 2.0 × 10^−6 a^	1.2 × 10^−5^ ± 1.2 × 10^−6 b^	2.4 × 10^−5^ ± 3.1 × 10^−7 a^
**Guadiamar**	1.2 × 10^−3^ ± 1.4 × 10^−9 a,b,c^	8.1 × 10^−6^ ± 1.1 × 10^−7 b^	1.8 × 10^−5^ ± 2.9 × 10^−6 a^	8.1 × 10^−6^ ± 6.6 × 10^−14 a,b^	1.6 × 10^−5^ ± 1.5 × 10^−7 a^
**Hispamar**	7.3 × 10^−4^ ± 5.3 × 10^−10 a^	5.1 × 10^−6^ ± 4.1 × 10^−7 a^	1.8 × 10^−5^ ± 1.8 × 10^−6 a^	5.1 × 10^−6^ ± 4.6 × 10^−7 a^	1.0 × 10^−5^ ± 9.8 × 10^−7 a^
**Jasmine**	1.4 × 10^−3^ ± 2.0 × 10^−9 b,c,d,e^	1.1 × 10^−5^ ± 1.7 × 10^−7 c,d^	1.8 × 10^−5^ ± 2.3 × 10^−6 a^	1.1 × 10^−5^ ± 1.2 × 10^−6 b^	2.2 × 10^−5^ ± 4.8 × 10^−13 a^
**Marisma**	1.3 × 10^−3^ ± 1.7 × 10^−9 a,b,c,d^	9.0 × 10^−6^ ± 8.0 × 10^−7 b,c^	1.8 × 10^−5^ ± 2.4 × 10^−6 a^	9.0 × 10^−6^ ± 1.4 × 10^−6 a,b^	1.8 × 10^−5^ ± 1.6 × 10^−6 a^
**Memby**	1.1 × 10^−3^ ± 1.2 × 10^−4 a,b^	8.7 × 10^−6^ ± 8.7 × 10^−9 b,c^	1.8 × 10^−5^ ± 1.5 × 10^−6 a^	8.7 × 10^−6^ ± 8.5 × 10^−7 a,b^	1.7 × 10^−5^ ± 1.9 × 10^−6 a^
**Perlado**	2.0 × 10^−3^ ± 2.4 × 10^−4 e,f^	1.1 × 10^−5^ ± 1.3 × 10^−7 c,d^	1.8 × 10^−5^ ± 1.4 × 10^−6 a^	1.1 × 10^−5^ ± 1.4 × 10^−6 b^	2.3 × 10^−5^ ± 3.7 × 10^−6 a^
**Piñana**	1.1 × 10^−3^ ± 1.0 × 10^−4 a,b^	7.9 × 10^−6^ ± 1.0 × 10^−7 b^	1.3 × 10^−4^ ± 1.2 × 10^−6 b^	7.9 × 10^−6^ ± 7.5 × 10^−7 a,b^	1.6 × 10^−5^ ± 1.6 × 10^−6 a^
**Puntal**	1.9 × 10^−3^ ± 1.9 × 10^−5 d,e,f^	1.3 × 10^−5^ ± 2.0 × 10^−7 d^	1.8 × 10^−5^ ± 2.0 × 10^−6 a^	1.3 × 10^−5^ ± 1.3 × 10^−7 b^	2.5 × 10^−5^ ± 3.2 × 10^−6 a^
**Sendra**	1.4 × 10^−3^ ± 1.4 × 10^−6 b,c,d,e^	1.2 × 10^−5^ ± 1.6 × 10^−7 d^	1.0 × 10^−5^ ± 2.9 × 10^−6 a^	1.2 × 10^−5^ ± 1.8 × 10^−7 b^	2.4 × 10^−5^ ± 3.2 × 10^−6 a^
**Sole**	1.6 × 10^−3^ ± 2.1 × 10^−9 b,c,d,e,f^	1.2 × 10^−5^ ± 9.6 × 10^−7 d^	1.8 × 10^−5^ ± 1.8 × 10^−6 a^	1.2 × 10^−5^ ± 1.3 × 10^−7 b^	1.8 × 10^−3^ ± 1.5 × 10^−4 b^
**Sona Masoori**	1.8 × 10^−3^ ± 2,0 × 10^−4 c,d,e,f^	1.3 × 10^−5^ ± 2.0 × 10^−7 d^	1.8 × 10^−5^ ± 2.3 × 10^−6 a^	1.3 × 10^−5^ ± 1.6 × 10^−6 b^	2.7 × 10^−5^ ± 2.1 × 10^−6 a^
**Thaiperla**	2.0 × 10^−3^ ± 2.4 × 10^−4 e,f^	8.7 × 10^−6^ ± 7.7 × 10^−7 b,c^	1.8 × 10^−5^ ± 2.4 × 10^−6 a^	8.7 × 10^−6^ ± 8.3 × 10^−7 a,b^	1.7 × 10^−5^ ± 1.7 × 10^−6 a^

Different letters in the same row indicate significant differences by ANOVA test (*p* < 0.05).

**Table 10 foods-10-02584-t010:** Percentage of metallic elements bioaccesibility (%).

	Al	Cd	Cr	Pb	As
**Barone**	59.41 ± 4.23 ^b,c,d,e,f^	47.99 ± 1.59 ^b,c^	34.85 ± 1.33 ^b,c^	47.90 ± 3.29 ^b,c^	36.71 ± 2.61 ^a,b^
**Basmati**	75.80 ± 3.05 ^e,f,g,h^	67.73 ± 1.99 ^e^	29.60 ± 1.93 ^a,b^	67.71 ± 2.24 ^e^	41.61 ± 4.25 ^a,b,c^
**Guadiamar**	52.47 ± 2.54 ^b,c,d^	45.12 ± 1.3 ^b,c^	44.47 ± 2.52 ^c,d^	45.12 ± 1.89 ^b^	36.28 ± 4.1 ^a,b^
**Hispamar**	28.43 ± 1.89 ^a^	28.60 ± 1.89 ^a^	76.95 ± 3.33 ^e^	28.63 ± 1.87 ^a^	50.16 ± 2.87 ^b,c,d^
**Jasmine**	67.95 ± 4.81 ^c,d,e,f,g,h^	59.97 ± 3.25 ^c,d,e^	33.42 ± 1.98 ^a,b,c^	59.97 ± 1.01 ^c,d,e^	31.61 ± 2.98 ^a^
**Marisma**	56.92 ± 4.01 ^b,c,d,e,^	50.18 ± 2.87 ^b,c,d^	49.93 ± 1.66 ^d^	50.11 ± 3.99 ^b,c,d,e^	66.20 ± 3.98 ^e^
**Memby**	65.54 ± 6.10 ^b,c,d,e,f,g^	48.41 ± 2.45 ^b,c,d^	51.64 ± 2.36 ^d^	48.42 ± 2.74 ^b,c^	66.24 ± 1.88 ^e^
**Perlado**	81.03 ± 4.54 ^g,h^	63.43 ± 2.96 ^d,e^	31.61 ± 2.98 ^a,b^	63.44 ± 1.79 ^d,e^	52.43 ± 2.54 ^c,d,e^
**Piñana**	59.67 ± 1.89 ^b,c,d,e,f^	43.19 ± 2.01 ^a,b^	54.80 ± 1.89 ^d^	43.96 ± 2.29 ^b^	43.99 ± 2.01 ^a,b,c^
**Puntal**	47.51 ± 1.99 ^a,b^	69.80 ± 1.25 ^e^	28.77 ± 0.98 ^a,b^	69.81 ± 1.55 ^e^	67.73 ± 1.99 ^e^
**Sendra**	72.02 ± 4.66^,d,e,f,g,h^	66.25 ± 3.98 ^e^	30.26 ± 1.57 ^a,b^	66.20 ± 1.88 ^e^	34.80 ± 1.33 ^a,b^
**Sole**	78.46 ± 2.29 ^f,g,h^	66.80 ± 1.96 ^e^	37.42 ± 1.64^,b^.^c^	66.88 ± 3.57 ^e^	59.94 ± 0.01 ^d,e^
**Sona Masoori**	87.60 ± 2.12 ^h^	74.41 ± 5.55 ^e^	22.45 ± 0.01 ^a^	74.45 ± 2.79 ^e^	47.92 ± 1.59 ^b,c,d^
**Thaiperla**	51.51 ± 2.22 ^b,c^	48.20 ± 1.98 ^b,c,d^	34.62 ± 0.99 ^b,c^	48.23 ± 3.31 ^b,c^	52.41 ± 2.54 ^c,d,e^

Different letters in the same row indicate significant differences by ANOVA test (*p* < 0.05).

**Table 11 foods-10-02584-t011:** Percentage of assimilated metallic elements respect to toxic threshold (%).

	Al	Cd	Cr	Pb	As
**Barone**	2.66 ± 0.19 ^a^	0.03 ± 0.001 ^a,b^	0.38 ± 0.0001 ^e^	0.03 ± 0.001 ^b^	0.07 ± 0.04 ^a^
**Basmati**	5.39 ± 0.29 ^b,c^	0.05 ± 0.001 ^c^	0.33 ± 0.001 ^d^	0.05 ± 0.002 ^d^	9.91 ± 4.45^,b^
**Guadiamar**	2.70 ± 0.98 ^a^	0.03 ± 0.0002 ^a,b^	0.31 ± 0.02 ^c^	0.03 ± 0.02 ^b^	0.06 ± 0.03 ^a^
**Hispamar**	2.41 ± 0.14 ^a^	0.02 ± 0.01 ^a^	0.02 ± 0.0001 ^a^	0.02 ± 0.001 ^a^	0.04 ± 0.03 ^a^
**Jasmine**	6.84 ± 0.97^c,d^	0.04 ± 0.002 ^b,c^	0.32 ± 0.02 ^c,d^	0.04 ± 0.002 ^c,^	0.08 ± 0.02 ^a^
**Marisma**	3.70 ± 0.21 ^a,b,c,d^	0.03 ± 0.001 ^a,b^	0.38 ± 0.0003 ^e^	0.03 ± 0.0001 ^b^	0.07 ± 0.03 ^a^
**Memby**	3.33 ± 0.16 ^a^	0.03 ± 0.001 ^a,b^	0.24 ± 0.001 ^b^	0.03 ± 0.002 ^b^	0.07 ± 0.02 ^a^
**Perlado**	5.26 ± 0.40 ^b,c^	0.04 ± 0.003 ^b,c^	0.25 ± 0.03 ^b^	0.04 ± 0.001^c^	0.09 ± 0.06 ^a^
**Piñana**	2.85 ± 0.2 ^a^	0.03 ± 0.002 ^a,b^	0.31 ± 0.001 ^c^	0.03 ± 0.002 ^b^	0.06 ± 0.02 ^a^
**Puntal**	5.67 ± 0.39 ^c^	0.05 ± 0.003 ^c^	0.83 ± 0.05 ^g^	0.05 ± 0.002 ^d^	0.10 ± 0.09 ^a^
**Sendra**	5.32 ± 0.35 ^b,c^	0.05 ± 0.004 ^c^	0.32 ± 0.002 ^c,d^	0.05 ± 0.001 ^d^	0.09 ± 0.08 ^a^
**Sole**	8.54 ± 0.65 ^d^	0.05 ± 0.003 ^c^	0.33 ± 0.002 ^d^	0.05 ± 0.002 ^d^	0.09 ± 0.08 ^a^
**Sona Masoori**	7.01 ± 0.48 ^c,d^	0.05 ± 0.002 ^c^	0.40 ± 0.003 ^f^	0.05 ± 0.01 ^d^	0.10 ± 0.06 ^a^
**Thaiperla**	3.58 ± 0.26 ^a,b^	0.03 ± 0.002 ^a,b^	0.24 ± 0.001 ^b^	0.03 ± 0.001 ^b^	0.07 ± 0.04 ^a^

Different letters in the same row indicate significant differences by ANOVA test (*p* < 0.05).

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
