# Peer review of "Nutritional Quality of the Most Consumed Varieties of Raw and Cooked Rice in Spain Submitted to an In Vitro Digestion Model"

_foods, 2021, doi:10.3390/foods10112584_

Round 1

Reviewer 1 Report

The article is interesting and brings new content to the current state of knowledge. The authors presented the results indicating the essence of their experiment. The introduction and abstract define which raw material was subjected to chemical analysis and for what purpose the experiment was carried out.
However, the authors do not discuss the obtained results with the existing data.
It is necessary to take into account the suggestions made in the notes to authors.

The title of the article fully reflects the subject and content. The authors formulated keywords that are closely related to the content of the article. These are important issues that will allow the reader to find the article in search engines.
The introduction is factual, but the reader should be alerted to the issues related to the purposefulness of the research undertaken. Thus, authors should provide specific reasons that guided the choice of the subject of the experiment and its relevance. This information is missing, and it is important because it highlights the current state of affairs.
The methodology is well described and provides comprehensive information on the course of the experiment. However, there was no statistical analysis of the results obtained. It is difficult to talk about significant differences when there are no such analyzes. It is a generally accepted principle that all research results, especially experimental ones, are subject to statistical analysis. The authors are asked to supplement the results with the results of the statistical analysis.
It is necessary to present and relate your experiment results to the available results of other authors. And also engage in discussions with them. The comparison of the selected results of the experiment based on the references and their interpretation is very important. Without discussing the results of the experiment, this is just a description of the results. Such a description is insufficient, especially since the authors describe the nutritional quality of the most commonly consumed raw and cooked rice varieties subjected to the in vito digestion model. Therefore, it becomes absolutely important to carefully discuss the obtained results in terms of the content of active substances determined in rice subjected to various processes, which so significantly modify its composition and the availability of active substances to the human body. The reader should be convinced of the purposefulness of the conducted research. It is impossible without discussion and references. And yet it seems that the authors undertook their actions for a specific purpose. Therefore, it is absolutely necessary to expand the description of the results with a substantive discussion based on the latest literature on the influence of the cooking and digestion process, which is now so well documented for rice.
The summary is laconic and does not contain any specific conclusions from the experiment. The authors are asked to indicate 2-3 conclusions that are important for the consumer and for the practice.

Author Response

The article is interesting and brings new content to the current state of knowledge. The authors presented the results indicating the essence of their experiment. The introduction and abstract define which raw material was subjected to chemical analysis and for what purpose the experiment was carried out.

However, the authors do not discuss the obtained results with the existing data.

It is necessary to take into account the suggestions made in the notes to authors.

The title of the article fully reflects the subject and content. The authors formulated keywords that are closely related to the content of the article. These are important issues that will allow the reader to find the article in search engines.

The introduction is factual, but the reader should be alerted to the issues related to the purposefulness of the research undertaken. Thus, authors should provide specific reasons that guided the choice of the subject of the experiment and its relevance. This information is missing, and it is important because it highlights the current state of affairs.

Following your suggestion this paragraph has been included at the end of the introduction section, in order to justify the choice of the subject and the relevance and originality of our study:

"There are several studies where the metallic composition of rice is studied but to our knowledge none of them are focused on the metallic and metalloid elements bioaccessibility of these wide varieties of rice."

The methodology is well described and provides comprehensive information on the course of the experiment. However, there was no statistical analysis of the results obtained. It is difficult to talk about significant differences when there are no such analyzes. It is a generally accepted principle that all research results, especially experimental ones, are subject to statistical analysis. The authors are asked to supplement the results with the results of the statistical analysis.

Ok you are totally right and all the statistical significance has been included in all the results presented in the tables (Table 6-11) and the figures 2-5.

It is necessary to present and relate your experiment results to the available results of other authors. And also engage in discussions with them. The comparison of the selected results of the experiment based on the references and their interpretation is very important. Without discussing the results of the experiment, this is just a description of the results. Such a description is insufficient, especially since the authors describe the nutritional quality of the most commonly consumed raw and cooked rice varieties subjected to the in vito digestion model. Therefore, it becomes absolutely important to carefully discuss the obtained results in terms of the content of active substances determined in rice subjected to various processes, which so significantly modify its composition and the availability of active substances to the human body.

Thank you for your advice. Following your recommendation, this paragraph has been included into the text.

 Page 8, section 3.2 “. However, all data obtained were similar to previously described by Vini G. et al with a mean protein value of 7.25±0.02%, and a mean carbohydrates value of 76.44±0.03% [38]. Indeed, the data are aligned with the results previously achieved by Yankah N., et al, with protein values of 7.89±0.16% and carbohydrate of 74.09±0.41% [39].”

The reader should be convinced of the purposefulness of the conducted research. It is impossible without discussion and references. And yet it seems that the authors undertook their actions for a specific purpose. Therefore, it is absolutely necessary to expand the description of the results with a substantive discussion based on the latest literature on the influence of the cooking and digestion process, which is now so well documented for rice.
Ok, thank you for your advice. Taking this suggestion into account different paragraphs have been added in the discussion section:

Page 8, section 3.2 "These results validate the ones previously described by Lim et al in 2003 and Ozbekova in 2019 [34,35]. In cooked rice moisture is much lower, showing an average of 4.09%±0.52, ranging between 5.2%±6.3 and 3.1%±4.2, found in the varieties Perlado and Thaiperla, respectively. However, it has not been possible to find preceding works that measured cooked rice moisture content by drying the rice before measuring. Indeed, values available in bibliography appear to be between 100-120%, for example in this study carried out by Wu et al. in 2017 [36]."

Page 10, section 3.3 " which is similar to mean values previously described by Rezvan A., et al (9.48±0.04 and 8,87±0.04%) in cooked Hashemi and Domsiyah varieties and the 9.74±0.21% found in cooked MGS variety by Kulwa F., et al [40,41]”

Page 15, section 3.4 " Even though this is the lowest value, it is substantially higher than the ones found by other authors. For instance, Kumari and Platel described a Cr bioaccessibility of 8 ± 0.17% [49]"

 Page 16 section 3.6 "However, there are some studies carried out in several countries that describe a THQ ex-ceeding the threshold limit, as it was reported by Hensawang., et al in 2017 in Thailand, Munish K., et al in India and Tapos K., et al in Bangladesh [52–54]”

The summary is laconic and does not contain any specific conclusions from the experiment. The authors are asked to indicate 2-3 conclusions that are important for the consumer and for the practice.

Ok, following your indications the abstract section has been modified and two new paragraphs have been included:

"Rice with a higher non-digestible fraction showed a higher liberation of proteins and a lower glycemic index. There were no significant differences in metallic and metalloid elements- concentrations in cooked rice or in digestible fraction in all varieties analysed."

"It has not been observed any relationship between the digestibility of rice and the bioaccessibility of each metallic and metalloid element. All of studied rice varieties are healthy food and its daily consumption is safe"

Reviewer 2 Report

In this manuscript, the nutritional quality, in terms of metals (Al, Cd, Pb, and Cr) and metalloid (As) content, of different varieties of raw/cooked rice was evaluated after in vitro digestion. The herein study is well-organized and it can be characterized as interesting and novel. However, there are some issues and considerations that must be addressed.

  • The authors must clearly indicate in the introduction the reason for the selection of these metals and metalloid. For example, the selection of Al is not at all discussed in this section. Please discuss this aspect.
  • The units in the manuscript are not consistent and/or does not correspond to the SI system and its abbreviations. For example, in line 86 please replace “grams” with “g” and in line 107 “hours” with “h”. Please make the required changes throughout text.
  • In section 2.5.2. the instrumentals parameters of the ICP method are presented. Did the authors develop the method, or did they use it from a previous work? If it was previously developed, the appropriate references must be provided. If they developed the method, how was its validity examined? Did any certified refence material was analyzed to proof method accuracy? These aspects must be also presented to indicate the accuracy of the obtained results.

Author Response

Answer to the referee 2

In this manuscript, the nutritional quality, in terms of metals (Al, Cd, Pb, and Cr) and metalloid (As) content, of different varieties of raw/cooked rice was evaluated after in vitro digestion. The herein study is well-organized and it can be characterized as interesting and novel. However, there are some issues and considerations that must be addressed.

The authors must clearly indicate in the introduction the reason for the selection of these metals and metalloid. For example, the selection of Al is not at all discussed in this section. Please discuss this aspect.

Ok, following your suggestion we have added new paragraphs in the introduction section in order to clarify the reason for the selection of these metals, from both toxicological and environmental points of view.

The units in the manuscript are not consistent and/or does not correspond to the SI system and its abbreviations. For example, in line 86 please replace “grams” with “g” and in line 107 “hours” with “h”. Please make the required changes throughout text.

Thank you very much for your advice. All the manuscript has been revised and all the units have been modified into the SI system.

In section 2.5.2. the instrumentals parameters of the ICP method are presented. Did the authors develop the method, or did they use it from a previous work? If it was previously developed, the appropriate references must be provided. If they developed the method, how was its validity examined? Did any certified refence material was analyzed to proof method accuracy? These aspects must be also presented to indicate the accuracy of the obtained results.

Page 4, section 2.5.2 “. These parameters were established and optimized for a liquid matrix previously by Gutierrez et al. [27].
